# Student-Led Campus Happiness Lunchboxes: Paying for Positive Impact

**Ze-Yung Wang [1] and Kuo-Wei Chen [2,*]**

[1] Department of Hospitality Management, Tajen University, Pingtung 90741, Taiwan; zyrudder@gmail.com
[2] Department of Hospitality Management, Ming Chuan University, Taoyuan 333326, Taiwan
* Correspondence: aa5433.zkhn@gmail.com

**Abstract:** In the aftermath of the 2022 Russian–Ukrainian war, Taiwan experienced economic shocks that prompted the government to initiate the happiness lunchbox program, aimed at fostering sustainable development and zero hunger concerns. Despite these efforts, economically disadvantaged university students faced challenges due to the unconventional outsourcing of campus meals. This study, conducted by leveraging campus culinary facilities, adopts an inferred value approach as opposed to the subjective willingness-to-pay (WTP) method, providing a more conservative assessment of students' willingness to contribute. Through regression analysis, this study highlights the positive correlation between student engagement in charitable activities and WTP for student-led events. This involvement not only enhances food safety and hygiene but also reflects a genuine commitment to supporting financially challenged students. The comprehensive nature of this approach effectively tackles issues related to campus nutrition, emphasizing the significance of establishing a sustainable campus environment to achieve objectives such as "zero hunger" and "responsible consumption and production" on campus.

**Keywords:** campus sustainability; zero hunger; economically disadvantaged individuals; food safety and hygiene

## 1. Introduction

The global commitment to sustainable development, as articulated in the United Nations' 2030 Agenda for Sustainable Development, underscores the urgency of addressing various challenges, with food security standing as a critical imperative among the 17 Sustainable Development Goals (SDGs) [1]. In alignment with the United Nations Sustainable Development Goal 12 (responsible consumption and production) and Sustainable Development Goal 2 (zero hunger), the Taiwan government has embarked on a mission to ensure food security, eliminate hunger, and promote sustainable agriculture [2]. This commitment has materialized through three key policies implemented by the Taiwan Council of Agriculture:

- Expanding services—filling the nutritional gap: By leveraging rural green care communities and agricultural cooperatives' green care stations, the government aims to expand meal services in terms of both capacity and coverage [3].
- Combatting food waste—stabilizing food supplies: The establishment of "food waste reduction zones" within agricultural cooperatives provides high-quality, affordable domestically produced ingredients [3].
- Public–private collaboration—extending care: The government has revised the "Domestic Food Assistance Operation Guidelines" to include social welfare organizations and green care stations, along with setting up a care hotline. This collaborative effort involves private enterprises, contributing to the provision of happiness boxes and food bundles for vulnerable populations, ensuring access to safe, nutritious, and sufficient food [3].

In 2015, the United Nations introduced the Sustainable Development Goals (SDGs), a global initiative aimed at addressing various socio-economic and environmental challenges. Simultaneously, recognizing the importance of fostering a sense of social responsibility within the education sector, Taiwan's Ministry of Education launched the "University Social Responsibility (USR) Program" in 2018. With an annual budget of approximately TWD 16.6 billion (equivalent to about USD 550 million), this initiative aims to promote sustainable practices and community engagement among universities [4–7]. As part of the USR Program, a dedicated fund of TWD 2.6 billion (around USD 86 million) is allocated to support universities in actively contributing to local communities and assisting under-privileged students. Over the past five years, collaborative efforts involving academia, industry, and government have led to heightened awareness among citizens and students regarding the United Nations' SDGs. This increased awareness has translated into a deeper understanding of Sustainable Development Goals, fostering a more proactive involvement in social responsibility and public welfare activities, particularly in the pursuit of SDG 2—zero hunger. This observed trend aligns with the findings of research conducted by Kopnina (2020) [8] and Manolis and Manoli (2021) [9], which underscore the positive impact of promoting Sustainable Development Goals on raising students' awareness and encouraging their active participation in socially responsible and public welfare initia-tives. Moreover, within the framework of Taiwan's Ministry of Education's USR plan, SDG 12—responsible consumption and production—has been incorporated. This inclusion reflects a commitment to cultivating sustainable consumption behavior and production models. Specifically, efforts are directed towards effectively reducing campus food waste and promoting the overall sustainable development of university campuses. The integra-tion of SDG 12 underscores the comprehensive approach adopted by Taiwan in aligning educational initiatives with global sustainability objectives.

The existing policies aimed at addressing food insecurity among university students in Taiwan have not been fully effective in supporting those who are economically dis-advantaged. This oversight leaves economically disadvantaged students exposed to the persistent issue of food insecurity. Furthermore, a significant contributing factor to this challenge is the operation of many university food services as outsourced ventures, creating an environment that often mirrors a seller's market. In this seller's market scenario, the prices of food products within campus eateries closely mirror market rates, and, in some instances, even exceed them. This pricing structure places an additional financial burden on economically disadvantaged university students, exacerbating their already challenging financial situation [5,10–13]. Beyond the economic implications, the outsourcing model adopted by universities has also led to concerns regarding the consistency of standards in food quality, safety, nutrition, and product pricing. The primary focus of food ser-vice providers on financial considerations raises questions about the overall safety and nutritional value of the food consumed by university students [14,15].

Addressing these challenges requires a comprehensive re-evaluation of existing poli-cies and practices to ensure that economically disadvantaged students receive adequate support in combating food insecurity. Moreover, it necessitates a closer examination of the outsourcing model to align with the broader goals of promoting student well-being, both financially and in terms of food safety and nutrition. The integration of these con-siderations into a cohesive and well-structured framework is imperative to create an envi-ronment where all university students can access affordable and nutritionally sound food options [5,6]. Addressing campus food security issues while integrating social responsibil-ity, ensuring an ample food supply, and providing reliable, healthy meals can significantly impact students' physical and mental well-being [16,17]. Within the framework of univer-sity sustainable development, SDG 12—sustainable consumption and production—plays a crucial role in reducing food loss and waste on campus, enhancing students' awareness of sustainable development, and contributing to the reduction of food waste and greenhouse gas emissions [18]. To achieve these goals, leveraging the capabilities of educational units within university hospitality management departments becomes essential. Fully equipped

cooking facilities in these departments can ensure food safety, balanced nutrition, and efficient ingredient procurement and management, surpassing the capabilities of typical small-scale food service providers. This presents a unique opportunity to instill social responsibility within educational institutions by utilizing the existing infrastructure to establish a community service mechanism. Through this mechanism, students can gain practical experience in meal preparation, production, and distribution.

Simultaneously, independent procurement, cooking, and marketing efforts empower students to develop self-management and teamwork skills. Operating an on-campus kitchen allows students to comprehend the intricacies of food preparation and safety, fostering a deeper appreciation for food security and the management of nutritious, delicious meals. Despite these potential benefits, there is a noticeable gap in previous academic research concerning student perceptions of student-led campus happiness lunchbox production, economic evaluations of such initiatives, and students' willingness to engage in socially responsible advocacy of zero hunger, promoting responsible campus consumption and production. This study aims to fill these gaps by building upon existing research conducted by previous scholars, contributing to a more comprehensive understanding of socially responsible measures to address zero hunger in campus settings. Its primary purpose is to enhance the campus community's collective efforts in solving the zero hunger problem through social responsibility initiatives. The study aspires to achieve a reduction in food loss and consumption on campus, propelling the campus towards sustainable development.

Recognizing the existing research gap, our study conducted a comprehensive survey to delve into students' willingness to pay (WTP) for campus-produced happiness lunchboxes and to evaluate their sense of social responsibility. We formulated a hypothesis acknowledging that relying solely on eliciting students' preferences through stated preference methods might introduce bias, particularly in terms of social desirability. To address this limitation, we expanded our inquiry to include students' inferred values, specifically their estimations of how much their peers would be willing to contribute. This additional measure served as a means to gauge students' donation intentions, providing a more holistic understanding of the factors influencing their willingness to donate and the associated amounts. By combining subjective WTP assessments with inquiries into students' perceptions of their peers' contributions, our study aimed to yield a more accurate assessment of students' attitudes and behaviors in the context of campus-produced happiness lunchboxes. This dual approach sought to minimize potential biases and enhance the reliability of the results, ultimately contributing to a more robust understanding of the factors shaping students' willingness to support socially responsible initiatives on campus.

## 2. Previous Studies and Theoretical Background

Stated Preference: Previous research has employed experimental surveys as a key methodology to quantify individuals' behavioral choices in novel contexts, as demonstrated by studies conducted by Kraft et al. (2022) and Tobi et al. (2019) [19,20]. These experimental surveys typically involve participants making choices among different strategies, products, or services characterized by varying ideal and adverse attributes [21]. For example, in a study focused on Generation Z, Narayanan (2022) utilized explicit preference surveys to reveal an increased concern for sustainability among participants. This heightened concern directly influenced their willingness to purchase and pay for corporate social responsibility initiatives [22]. As such, stated preference methods have emerged as the preferred approach for estimating the value of non-market services, particularly those associated with on-campus social responsibility initiatives, such as the creation and distribution of happiness lunchboxes [23,24].

Research highlights a pervasive limitation in general social surveys known as "social desirability bias," wherein respondents may provide inaccurate responses influenced by societal expectations [25]. Lopez-Becerra and Alcon (2021) demonstrated that consumers tend to overstate their commitment to protecting the natural environment and inflate their

spending on green products due to this bias. This bias, if unaddressed, poses a significant challenge in understanding students' engagement in campus social responsibility-related measures [26]. The assumed nature of students' service may be distorted, casting doubt on the validity of assessment results. Without implementing techniques or methods to mitigate social desirability bias, accurate evaluations of students' contributions to such initiatives become questionable.

In the context of this study, it is crucial to emphasize that social responsibility lacks readily available market prices for valuation, making its assessment challenging. Therefore, drawing insights from previous research, we have identified the contingent valuation method as the most suitable approach [27,28]. The contingent valuation method entails directly querying individuals about their willingness to pay (WTP) for a specific target, which, in this study, pertains to the value of social responsibility initiatives. Nevertheless, it is imperative to acknowledge the limitations of the contingent valuation method, as highlighted by Yadav et al. (2022) and Bostan et al. (2020), especially when estimating the value of public goods or similar non-market transactions without direct or indirect market prices [29,30]. To address potential biases introduced by factors such as social desirability and to enhance the precision of our assessment, this study adopts the inferred valuation method. This approach has been validated in the research conducted by Lopez-Becerra and Alcon (2021) and Sakurai and Uehara (2023) [26,31]. The inferred valuation method provides a more accurate evaluation, ensuring a robust foundation for our examination of the value associated with social responsibility initiatives.

Inferred Valuation Method: Grounded in the understanding of how individuals anticipate the behavior of others to enhance the prediction of collective group behavior, the inferred valuation method serves as a valuable tool in mitigating social desirability bias [26,32]. As recommended by Lusk and Norwood (2009), this method enables respondents to consider social expectations without compromising their own sense of pleasure or happiness [33]. Extensive research by Lopez-Becerra and Alcon (2021) has further validated the effectiveness of the inferred valuation method in minimizing the disparity between respondents' actual behavior and their predicted behavior [26]. Our study relies on the inferred valuation method, recognizing its ability not only to align respondents' behavior more closely with their inferred values but also to offer a more precise estimation of our study's scale in real-world conditions. Past research underscores the superiority of the inferred valuation method over subjective willingness-to-pay (WTP) approaches in delivering a more realistic and precise assessment. Lopez-Becerra and Alcon (2021) emphasize the impact of social expectation bias on traditional estimates in non-market valuation methods, emphasizing the need for validation through the inferred valuation method [26]. In addressing biases in individuals' willingness to pay, Lusk and Norwood (2009) [33] argue that the inferred valuation method effectively examines individuals' perceptions of a product's value based on their predictions or inferences of others' values, rather than relying on their own potentially biased values. This approach contrasts with traditional stated assessment values, as demonstrated by Yadav et al. (2022) [29], who found significant disparities between inferred values and traditional estimates in a preference study in Ireland, with inferred values being only a third of the traditional estimates [29,33]. Consequently, the inferred valuation method yields a more conservative and realistic estimation in our study.

However, despite numerous studies focusing on inferred valuation, there remains a dearth of research specifically examining students' inferred values in relation to the development of sustainable campus meal boxes through active participation in social responsibility initiatives. Evaluating the actual costs associated with social responsibility, encompassing ingredient procurement, production, and sales within the context of these meal boxes, is crucial for decision makers to assess the viability of such projects. Previous research has highlighted that university students are not only willing but eager to contribute financially to social responsibility initiatives, with their total willingness-to-pay (WTP) estimates consistently exceeding the actual implementation costs [6,7,34,35]. Furthermore, the promotion of university social responsibility not only enhances the institution's brand

image but also nurtures a sense of social responsibility and community engagement among students [6,34,36]. If the expenses associated with establishing an on-campus meal box production facility can be covered by student donations, these initiatives can be sustained without heavy reliance on external funding from social welfare programs or dedicated campus funds aimed at supporting economically disadvantaged students. Additionally, supporting campus happiness lunchboxes aligns with the United Nations' Sustainable Development Goals (SDGs), specifically SDG12 (responsible consumption and production) and SDG2 (zero hunger). This endeavor serves as a positive demonstration of moral education for students, fostering a commitment to fulfilling their social responsibilities. Importantly, it presents a practical and viable approach for achieving sustainable campus development [5–9,37]. Therefore, it is crucial to recognize that social expectations may lead to an overestimation of the value of such social responsibility initiatives for students. Consequently, utilizing the inferred valuation method to explore the authentic value of students' willingness to pay is a validated and effective approach.

The global economic repercussions of the COVID-19 pandemic have indirectly impacted the lives of students. The economic downturn resulting from the pandemic has led to a rise in unemployment rates among students' parents or a reduction in their income, thereby diminishing their financial capacity to support their children [4,37]. Consequently, students have experienced a reduction in disposable income, prompting changes in their spending habits and placing some students in a precarious situation where meeting basic needs becomes challenging [5,6,10,14]. It is noteworthy that an increase in financial support from students' parents may raise their standards for material assistance [38]. As students become more financially secure, their expectations of social welfare goods may also elevate, potentially reducing their inclination to purchase such products [39]. This shift in perspective may be influenced by the development of students' attitudes toward a sense of social responsibility and public giving, resembling a form of purchasing "Indulgences" to fulfill societal obligations without direct personal engagement [40]. However, it is essential to acknowledge the dynamic nature of socio-economic contexts, which may lead respondents to exhibit variability in evaluating the value of a service. Therefore, this study aims to assess students' values across different years (e.g., freshmen in 2021 and 2022) to determine if variations exist in students' value assessments. Building upon the aforementioned research, the first hypothesis of this study posits that the inferred willingness to pay (WTP) for student donations of happiness lunchboxes on student-led campuses will be smaller than the students' own subjective WTP. The second hypothesis suggests that there are differences in donation-inference WTP and subjective WTP among students enrolled in different academic years.

## 3. Calculation

In the context of this study, the estimation calculation builds upon the model proposed by Levitt and List (2007) with some refinements [41]. The assessment employs a utility function approach, as suggested by Lusk and Norwood (2009) [33].

$$U = w^{NH} M (A,H) + (1 − w^{NH}) V (I − A,E)$$

After organizing the above formulas:

M: The utility obtained through social norm behavior
A: Individual behavior after adopting social norm expectations (A = $WTP^{NH}$).
H: The integrity level of the respondents
V: General utility
I: Respondents' income
E: The value of campus social responsibility in this study
W: Weighted score representing the relationship between social responsibility and consumption

The socially desired behavior in this study is a campus happiness lunchboxes donation payment, where we consider actual (not hypothetical) willingness to pay ($WTP^{NH}$). When willing to pay for positive donations, $A = WTP^{NH}$. If there is no willingness to donate, $A = 0$. H represents the individual's degree of honesty. As the difference between WTP and the actual payment amount increases, H will decrease. On the contrary, when the actual payment amount is equal to WTP, $H = 0$.

When respondents make donations, their income decreases ($I-WTP^{NH}$), and the value of social responsibility increases ($E\to E'$). Note that $w^{NH}$ is a weight that determines the relative importance of $M(\cdot)$ and $V(\cdot)$ ($0 \le wNH \le 1$). The existence of $M(\cdot)$ not only shows that personal utility is not only affected by income and enjoyment of environmental benefits, but also by the donation behavior itself. This is because improving the campus atmosphere through happiness lunchboxes is a desirable behavior. Therefore, when practicing ethical behavior through a donation, there will be a certain sense of satisfaction. Therefore, the impact of $M(\cdot)$ on utility will be defined by $w^{NH}$. The larger the $w^{NH}$, the greater the impact on utility.

When student-driven donations for social responsibility activities are perceived as socially expected behavior, the value of M is $w^{NH}$ increases. Therefore, it is assumed that student-driven social responsibility activities and donations are hypothetically equivalent, hence NH (non-hypothetical) equals H (hypothetical). Based on this theoretical analysis, it can be asserted that both a willingness to donate and integrity contribute positively to utility ($MA > 0$, $MH > 0$). Consequently, it is hypothesized that $WTP^{H}$ will be greater than $WTP^{NH}$.

## 4. Methods

This study was conducted within the Tourism College of a major university, encompassing students from various fields, including Tourism Management and Hospitality Management, among others. As students from different departments were involved, their interests spanned multiple domains. The first survey was administered to a portion of incoming freshmen in September 2021 ($n = 135$), while the second survey was conducted with a separate group of incoming freshmen in September 2022 ($n = 143$).

Freshmen enrolling in September 2021 completed their high school education between September 2018 and June 2021, while freshmen enrolling in September 2022 completed their high school education between September 2019 and June 2022. The Ministry of Education's Higher Education Intensification Program in Taiwan encourages collaboration between colleges and universities to guide first-year high school students in engaging in meaningful social responsibility initiatives. These projects are exclusively implemented during the initial year of high school, requiring all first-year students to actively participate and obtain a study certificate. Consequently, students admitted in 2021 were involved in the social responsibility practice plan of the Higher Education Deepening Program throughout their high school journey. However, the 2022 entrants face unique challenges as they coincided with the peak period of the COVID-19 pandemic (October 2019 to December 2022). As a result, the relevant social responsibility practice plans for these students were provisionally scheduled, with a significant portion being conducted through remote online teaching. This situation has led to the 2022 freshmen not receiving a comprehensive social responsibility practice plan during their high school tenure. Additionally, these newly admitted students have not fully assimilated into the cultural and aspirational fabric of the university or college, rendering them a distinct student cohort.

This study's questionnaire is crafted based on an exhaustive compilation of past academic theories and literature. Three experts and scholars in related fields, representing academic, social welfare, and governmental perspectives, reviewed the questions for validity. The process includes expert interviews, sequential analysis, and item analysis, ensuring a rigorous pursuit of higher reliability. Factor extraction was performed through exploratory factor analysis, utilizing the maximum variation method for orthogonal rotation on the component matrix, resulting in the identification of three factors. The final version

of the questionnaire encompasses three dimensions: interest in social responsibility and the creation of happiness lunchboxes, engagement in social responsibility and philanthropic behaviors, and personal attributes of the respondents. The Likert five-point scale was employed, ranging from "1" (strongly disagree) to "5" (strongly agree), as illustrated in Table 1.

**Table 1.** Questionnaire items and reliability validity analysis.

| Dimension | Question Item | Factor Loading | Cronbach's α |
|---|---|---|---|
| I. Interest in social responsibility and the production of happiness lunchboxes. | I1. I am interested in social responsibility. | 0.733 | 0.819 |
| | I2. I am interested in giving back to society through public welfare. | 0.691 | |
| | I3. I am interested in producing happiness lunchboxes on campus. | 0.725 | |
| | I4. I am interested in the process of meal preparation. | 0.692 | |
| | I5. I am interested in eating happiness lunchboxes made by myself. | 0.786 | |
| | I6. I am interested in eating balanced diet happiness lunchboxes. | 0.759 | |
| B. Socially responsible charitable acts. | B1. I will strive to engage in social responsibility to the best of my abilities. | 0.889 | 0.864 |
| | B2. I will try my best to purchase socially responsible and charitable products. | 0.891 | |
| | B3. I am willing to contribute to social welfare through donations. | 0.881 | |
| F. Characteristics of the interviewee. | F1. Age. | 1. 17 year old  2. 18 years old  3. 19 years old  4. 20 years old  5. Over 21 years old | |
| | F2. I have a part-time job to earn income. | Dichotomies | |
| | F3. My parents provide financial support for my living expenses and allowances. | Dichotomies | |

The questionnaire used in this study elucidated the research's content and purpose and affirmed that student participation was voluntary. Once students agreed to the questionnaire's terms, they proceeded to answer the subsequent questions. The questionnaire encompassed three dimensions: interest in social responsibility and the production of happiness lunchboxes, engagement in social responsibility and philanthropic behaviors, and personal attributes of the respondents. Regarding the investigation of WTP, this study employed a binary choice format. Respondents were presented with donation amounts and were asked to either accept or decline. To enhance the sample size effectively, a double-bounded dichotomous choice approach was used. After respondents provided their initial responses, we attempted to increase or decrease the price and asked whether they would accept or decline the donation amount a second time. The study presented five different donation amounts for assessment A: TWD 100 (about USD 3.3), B: TWD 200 (about USD 6.6), C: TWD 400 (about USD 13.2), D: TWD 800 (about USD 26.4), E: TWD 1600 (about USD 52.8).

During the interview process, respondents were provided with a comprehensive understanding of the hypothetical scenario (establishing a student-led on-campus happiness lunchboxes catering initiative) and informed about the actual involvement in social responsibility activities, as shown in Table 2. Respondents were made aware that the happiness lunchboxes not only catered to economically disadvantaged students on campus but also

allowed participants to freely donate and purchase the nutritionally balanced, healthy, and delicious meal boxes at the same economic price. After this thorough explanation, respondents were asked if they were willing to participate in social responsibility and make a donation. Furthermore, after inquiring about the respondents' willingness to donate, the study employed the inferred valuation method to inquire, "Do your friends or acquaintances at school also wish to donate?" This approach further mitigated the respondents' social desirability bias, enabling a more realistic and accurate assessment of their WTP.

**Table 2.** Scenarios presented in the investigation.

| |
|---|
| The on-campus catering and hospitality cooking class is a lunchbox production kitchen that complies with food safety and hygiene standards, hereinafter referred to as the (lunchbox kitchen). We are trying to consider promoting campus happiness lunchboxes, and the campus happiness lunchboxes team led by the newly established student group will produce campus happiness lunchboxes. (This is a hypothetical issue and will be implemented when conditions permit). Students who are interested in sustainable development and social responsibility on campus can participate in this activity. In addition, the campus happiness lunchboxes are provided free of charge to disadvantaged students on campus. Faculty and staff who want to eat healthy and delicious lunchboxes can also be provided for a fee. |
| In order to produce happiness lunchboxes, it is necessary to prepare staple foods, ingredients, seasonings, packaging containers, etc. Initially, the production costs of the campus happiness lunchboxes, which are run under the leadership of student groups, will be donated through donations from students who are willing to support the implementation of the activity. If, without sufficient donations, the student-led happiness lunchboxes Team will be unable to produce the happiness lunchboxes kitchen, it will be returned to teaching use. |
| If I would like you to donate TWD XXXX (as a one-time donation for the entire semester), would you be willing to donate? Please note that the amount of your donation will directly affect your allocation of other funds. |

(Each survey shows a different bid amount [Survey A: TWD 100; Survey B: TWD 200; Survey C: TWD 400; Survey D: TWD 800; Survey E: TWD $1600]).

## 5. Analysis

To begin, it is imperative to ascertain significant differences in attributes and cognitive factors between the two groups. Therefore, chi-square tests and independent *t*-tests were initially employed. The results of these tests all indicated that there was independence between the two groups, and statistically significant differences existed, as shown in Tables 3 and 4.

**Table 3.** Various statistics of student groups.

| Dimension | Enrollment Year | N | Mean | Std. Deviation | Std. Error Mean |
|:---:|:---:|:---:|:---:|:---:|:---:|
| I | 2021 | 135 | 4.022 | 0.621 | 0.054 |
| | 2022 | 146 | 3.352 | 0.923 | 0.077 |
| B | 2021 | 135 | 3.165 | 0.809 | 0.070 |
| | 2022 | 146 | 2.909 | 0.745 | 0.062 |

I: Interest in social responsibility and the production of happiness lunchboxes. B: Socially responsible charitable acts.

**Table 4.** Independent sample test of student groups.

| Dimension | *t*-Test for Equality of Means | | | | | | |
|:---:|:---:|:---:|:---:|:---:|:---:|:---:|:---:|
| | T | df | Sig. (2-Tailed) | Mean Difference | Std. Error Difference | 95% Confidence Interval of the Difference | |
| | | | | | | Lower | Upper |
| I | 7.140 | 250.016 | 0.000 | 0.670 | 0.094 | 0.486 | 0.855 |
| B | 2.751 | 0.276 | 0.006 | 0.256 | 0.094 | 0.073 | 0.440 |

I: Interest in social responsibility and the production of happiness lunchboxes. B: Socially responsible charitable acts.

Following the execution of the independent sample *t*-test, the obtained results are presented in Table 4. The findings reveal a notable distinction in Dimension: I (interest

in social responsibility and the production of happiness lunchboxes) between students enrolled in 2021 and 2022. According to the independent sample *t*-test table, the t-value is 7.140, with a significance level of 0.000 (<0.05). Subsequent to the post-test analysis, it was discerned that the average value of Dimension: I for students enrolled in 2021 exceeds that of students enrolled in 2022. Additionally, a significant disparity is observed in Dimension: B (socially responsible charitable acts) between students admitted in 2021 and 2022. Based on the independent sample *t*-test table, the t-value is 2.751, and the significance is 0.006 (<0.05). Post-test results indicate that the average Dimension: B of students admitted in 2021 surpasses that of students admitted in 2022. Consequently, it can be inferred that a substantial difference exists between Dimension: I and Dimension: B among students enrolled in 2021 and 2022, both demonstrating independence. Subsequently, this study utilized students' donation amounts as the independent variable and conducted an analysis using a simple logit model to understand students' willingness to pay (WTP) and estimate the value of student-led social responsibility activities in producing happiness lunchboxes for the 2021 and 2022 student samples. The simple logit model assumes utility differences as follows:

$$\text{Constant} + \text{Coefficient} \times \log_e(Donation) + \text{Error}$$

Thus, the agreement rate is represented as $1/(1 + \exp(-\Delta V))$, where $-\Delta V$ is the sum of the fixed errors derived from the utility difference. The log-likelihood equation is as follows:

$$\log_e L = \sum_{i=1}^{n} \left[ D_i^{AA} \log_e \pi_{AA}\left(A, A^U\right) + D_i^{AD} \log_e \pi_{AD}\left(A, A^U\right) \right.$$
$$\left. + D_i^{DA} \log_e \pi_{DA}\left(A, A^L\right) + D_i^{DD} \log_e \pi_{DD}\left(A, A^L\right) \right]$$

where *n* represents the observable variables. *D* is a dummy variable, with $^{AA}$ denoting agreement to donate in both the first and second rounds, $^{AD}$ indicating agreement to donate in the first round but not in the second, $^{DA}$ indicating disagreement to donate in the first round but agreement in the second, and $^{DD}$ representing disagreement in both rounds. A signifies the first donation amount, and if the respondent agrees, $A^U$ represents the second donation amount, which is higher than the first. Conversely, if the respondent disagrees with the first donation amount, $A^L$ represents the second donation amount, which is lower than the first. This study will be conducted at a 95% confidence interval to ensure statistical significance.

Furthermore, this study will employ a simple logit model for model regression to investigate factors affecting WTP and inferred values. Therefore, both the 2021 and 2022 samples will be integrated into a comprehensive model regression for analysis, including all variables as independent variables. These independent variables comprise six factors related to interest in social responsibility and happiness lunchboxes production, three factors related to social responsibility and philanthropic behaviors, and three factors related to sample attributes. Additionally, the model incorporates variables such as the year of the respondent and the log (*donation*) amount, resulting in a total of 15 factors as independent variables. Thus, the function [$V_{ij}$] formed for each individual i and alternative j in this study is as follows:

$$V_{ij} = b_0 + b_1 F1 + b_2 F2 + b_3 F3 + b_4 I1 + b_5 I2 + b_6 I3 + b_7 I4 + b_8 I5 + b_9 I6 + b_{10} B1 + b_{11} B2$$
$$+ b_{12} B3 + b_{13} EY + b_{14} \log(donation) + e$$

where $b_0$ is the intercept. $b_1$, $b_2$, and $b_3$ represent coefficients for respondent characteristics, including age (F1), part-time work (F2), and parental financial support (F3). $b_4$, $b_5$, $b_6$, $b_7$, $b_8$, and $b_9$ correspond to coefficients for factors related to interest in social responsibility and happiness lunchboxes production, such as interest in social responsibility (I1), engagement in social philanthropy (I2), involvement in on-campus happiness lunchboxes production (I3), interest in the meal preparation process (I4), satisfaction from consuming self-produced meal boxes (I5), and preference for a balanced diet meal box (I6), etc. $b_{10}$, $b_{11}$, and $b_{12}$ are coefficients for social responsibility and philanthropic behaviors, including involvement in

social responsibility initiatives (B1), willingness to purchase socially responsible products (B2), and willingness to donate to social philanthropy (B3), etc. $b_{13}$ represents the coefficient for the respondent's year of enrollment. $b_{14}$ is the coefficient for the log of the donation amount log (*donation*). e denotes the error term.

Statistical analysis was performed using SPSS 25.0, which included descriptive statistics, chi-square tests, independent *t*-tests, correlation coefficient (r) calculations for WTP and inferred values, and comprehensive model regression analysis.

## 6. Results

In the academic year 2021, 82.2% of enrolled students were above the age of 18, while in 2022, this proportion experienced a slight decline to 79.0%. In terms of engaging in part-time employment for income generation, the involvement of students increased significantly from 60.00% in 2021 to 81.82% in 2022. Financial support from parents exhibited a contrasting trend, with 65.19% of 2021 students receiving such assistance, whereas this percentage decreased notably to 39.86% for students in 2022.

The chi-square test results presented in Table 5 highlight a significant difference in part-time employment between students enrolled in 2021 and 2022 ($p < 0.01$). Specifically, a majority of students in both years are engaged in part-time jobs, emphasizing the noteworthy increase in part-time employment among 2022 students. Furthermore, the chi-square test underscores a significant disparity in financial support from parents, with a higher proportion of 2021 students receiving such support compared to their 2022 counterparts ($p < 0.01$).

In evaluating students' interest in Dimension: I (interest in social responsibility and the production of happiness lunchboxes), it is observed that, while students in both admission years express a general interest, those admitted in 2021 exhibit a significantly higher interest in specific items within this dimension ($p < 0.01$). Notably, students admitted in 2021 show a heightened interest in Item: I1 (I am interested in social responsibility), Item: I2 (I am interested in giving back to society through public welfare), Item: I3 (I am interested in producing happiness lunchboxes on campus), Item: I4 (I am interested in the process of meal preparation), Item: I5 (I am interested in eating happiness lunchboxes made by myself), and Item: I6 (I am interested in eating a balanced diet in happiness lunchboxes).

In the context of Dimension: B (socially responsible charitable acts), students admitted in 2021 respond more positively to Item: B1 (I will strive to engage in social responsibility to the best of my abilities) compared to their 2022 counterparts ($p < 0.05$). Additionally, students enrolling in 2021 express a greater inclination towards Item: B3 (I am willing to contribute to social welfare through donations) than students enrolling in 2022, demonstrating a significant difference ($p < 0.01$), as outlined in Table 6.

**Table 5.** Part-time student employment and financial support from parents.

| Items | Actual Situation | Entry Year | | Chi-Square Score | *p* Value | Significance |
|---|---|---|---|---|---|---|
| | | 2021 (*n* = 135) | 2022 (*n* = 143) | | | |
| Part-time Student Employment | No | 54 | 26 | 16.129 | 0.000 | *** |
| | Yes | 81 | 117 | | | |
| Financial Support from Parents | No | 47 | 86 | 17.848 | 0.000 | *** |
| | Yes | 88 | 57 | | | |

Variables of Significance (*** $p \leq 0.001$).

**Table 6.** Students' interest and behavior regarding social responsibility.

| Item | 2021 (*n* = 135) | | 2022 (*n* = 143) | | T | *p* |
|---|---|---|---|---|---|---|
| | **Mean** | **SD** | **Mean** | **SD** | | |
| I1. I am interested in social responsibility. | 4.51 | 0.645 | 4.17 | 0.959 | 3.448 | 0.001 |
| I2. I am interested in giving back to society through public welfare. | 3.69 | 1.136 | 3.01 | 1.345 | 4.553 | 0.000 |
| I3. I am interested in producing happiness lunchboxes on campus. | 3.90 | 1.190 | 3.43 | 1.202 | 3.275 | 0.001 |
| I4. I am interested in the process of meal preparation. | 3.65 | 1.155 | 2.78 | 1.369 | 5.729 | 0.000 |
| I5. I am interested in eating happiness lunchboxes made by myself. | 4.24 | 0.918 | 3.46 | 1.197 | 6.139 | 0.000 |
| I6. I am interested in eating balanced diet happiness lunchboxes. | 4.13 | 0.904 | 3.25 | 1.313 | 6.549 | 0.000 |
| B1. I will strive to engage in social responsibility to the best of my abilities. | 3.22 | 0.895 | 3.01 | 0.852 | 2.055 | 0.041 |
| B2. I will try my best to purchase socially responsible and charitable products. | 3.09 | 0.942 | 2.94 | 0.841 | 1.415 | 0.158 |
| B3. I am willing to contribute to social welfare through donations. | 3.19 | 0.883 | 2.78 | 0.849 | 3.871 | 0.000 |

This study conducted a detailed analysis of the willingness to donate among students who expressed a readiness to contribute, excluding those who did not. The average subjective willingness to pay (WTP) among students admitted in 2021, based on the highest bid, was found to be TWD 458.43 (approximately USD 15.03) (*n* = 89; Std. Deviation = 369.831). In comparison, the inferred average WTP was TWD 417.24 (approximately USD 13.68) (*n* = 58; Std. Deviation = 374.699). Consequently, the subjective WTP for donation willingness among students enrolled in 2021 was TWD 41.19 (approximately USD 1.35) higher than the inferred WTP. However, these differences were deemed statistically insignificant (*p* = 0.513) due to the overlapping values of subjective WTP and inferred WTP.

For students entering in 2022, the average subjective WTP, based on the highest bid, was TWD $327.50 (approximately USD $10.73) (*n* = 80; Std. Deviation = 191.579). In contrast, the inferred average WTP was TWD $377.36 (approximately USD $12.37) (*n* = 83; Std. Deviation = 272.906). Notably, the subjective WTP of students enrolling in 2022 was $49.86 (approximately USD $1.63) lower than the inferred WTP. Nevertheless, similar to the 2021 cohort, these differences were not statistically significant (*p* = 0.218) as the subjective WTP and inferred WTP values exhibited overlap, as illustrated in Table 7.

**Table 7.** Independent sample test of subjective WTP and inferred WTP.

| Year | WTP | N | Mean | SD | *t*-Test for Equality of Means | | | | | | |
|---|---|---|---|---|---|---|---|---|---|---|---|
| | | | | | T | df | Sig. (2-Tailed) | Mean Difference | Std. Error Difference | 95% Confidence Interval of the Difference | |
| | | | | | | | | | | Lower | Upper |
| 2021 | S | 89 | 458.43 | 369.831 | 0.657 | 145 | 0.513 | 41.186 | 62.734 | −82.806 | 165.177 |
| | I | 58 | 417.24 | 374.699 | | | | | | | |
| 2022 | S | 80 | 327.50 | 191.579 | −1.238 | 131 | 0.218 | −49.858 | 40.270 | −129.521 | 29.804 |
| | I | 53 | 377.36 | 272.906 | | | | | | | |

S: Subjective WTP. I: Inferred WTP.

The outcomes of the regression analysis in the current study revealed notable insights into the factors influencing students' subjective willingness to pay (WTP). Specifically, six independent variables demonstrated significant impacts on the students' inclination to contribute a higher subjective WTP. These variables are as follows: I2—interest in giving back to society through public welfare undertakings (B = 0.153, *p* < 0.01); I3—interest in creating happiness lunchboxes on campus (B = 0.255, *p* < 0.05); I4—interest in the cooking process (B = 0.175, *p* < 0.05); I6—interest in balanced diet meal boxes (B = 0.189, *p* < 0.05);

F2—engagement in part-time employment for income (B = 0.718, $p < 0.01$); and F3—financial support from parents (B = 1.176, $p < 0.01$). The calculated average WTP was determined to be TWD 396 (approximately USD 13.05).

Conversely, the results pertaining to inferred WTP also highlighted six influential independent variables. These variables significantly influenced students to express higher inferred WTP. The identified variables are as follows: I2—interest in contributing to society through public welfare undertakings (B = 0.147, $p < 0.05$); B1—efforts to assume social responsibilities (B = 0.579, $p < 0.01$); B2—willingness to purchase socially responsible products (B = 0.359, $p < 0.01$); B3—voluntary contributions to social services through donations (B = 0.537, $p < 0.01$); F2—involvement in part-time employment for income (B = 0.768, $p < 0.01$); and F3—receipt of financial support from parents (B = 0.642, $p < 0.01$). The inferred average WTP was calculated to be TWD 398 (approximately USD 13.05).

Comparing the subjective WTP model with the inferred WTP model, it was observed that both models collectively influence the students' willingness to donate. Common factors impacting both models include—I2: interest in giving back to society through public welfare undertakings; F2: earning income from part-time jobs; and F3: parents providing financial support (refer to Table 8 for details).

**Table 8.** Interest and behavior of students in social responsibility regarding Subjective WTP and Inferred WTP.

| Item | Subjective WTP (*n* = 169) | | | Inferred WTP (*n* = 111) | | |
|---|---|---|---|---|---|---|
| | B | T | P | B | T | P |
| Intercept | −2.381 | −2.796 | 0.006 ** | −7.567 | −7.022 | 0.000 *** |
| F1. Age. | 0.023 | 0.530 | 0.597 | −0.024 | −0.539 | 0.591 |
| F2. I have a part-time job to earn income. | 0.718 | 4.489 | 0.000 *** | 0.768 | 4.100 | 0.000 *** |
| F3. My parents provide financial support for my living expenses and allowances. | 1.176 | 8.540 | 0.000 *** | 0.642 | 4.865 | 0.000 *** |
| I1. I am interested in social responsibility. | 0.134 | 1.347 | 0.180 | 0.179 | 1.869 | 0.065 |
| I2. I am interested in giving back to society through public welfare. | 0.153 | 2.387 | 0.018 * | 0.147 | 2.008 | 0.047 * |
| I3. I am interested in producing happiness lunchboxes on campus. | 0.255 | 3.327 | 0.001 ** | 0.092 | 1.126 | 0.263 |
| I4. I am interested in the process of meal preparation. | 0.175 | 2.583 | 0.011 * | 0.082 | 1.071 | 0.287 |
| I5. I am interested in eating happiness lunchboxes made by myself. | 0.120 | 1.625 | 0.106 | 0.094 | 1.197 | 0.234 |
| I6. I am interested in eating balanced diet happiness lunchboxes. | 0.189 | 2.626 | 0.010 * | 0.032 | 0.413 | 0.681 |
| B1. I will strive to engage in social responsibility to the best of my abilities. | −0.038 | −0.361 | 0.719 | 0.579 | 6.216 | 0.000 *** |
| B2. I will try my best to purchase socially responsible and charitable products. | −0.087 | −0.894 | 0.373 | 0.359 | 4.024 | 0.000 *** |
| B3. I am willing to contribute to social welfare through donations. | −0.079 | −0.727 | 0.468 | 0.537 | 5.456 | 0.000 *** |
| Entry Year | −0.011 | −0.100 | 0.920 | 0.396 | 3.918 | 0.000 *** |

Variables of Significance (* $p \leq 0.05$, ** $p \leq 0.01$, *** $p \leq 0.001$).

This study surveyed about 1200 students in the School of Tourism and used the sample to calculate the average inferred WTP. For the student-led social responsibility activity of making happiness lunchboxes on campus, the estimated utility value is as follows:

$$398 \times 1200 \times 0.6079 = \text{TWD } 290{,}333 \text{ (about USD 9519.1)}$$

(Note: The conversion rate used here is approximate and may vary with exchange rates).

## 7. Discussion

### 7.1. Cognitive and Demographic Comparisons between Students Enrolled in 2021 and 2022

This research conducted a comprehensive survey of incoming students from various academic years, with a particular focus on the cohorts entering in 2021 and 2022 within the same university and college. The findings revealed discernible differences in their attitudes towards social responsibility, engagement in socially responsible behaviors, participation levels, and certain demographic characteristics. It was observed that students entering in 2021 demonstrated a heightened interest in social responsibility compared to their counterparts in 2022. However, intriguingly, their involvement in socially beneficial activities, such as the creation of happiness lunchboxes, showed a lower level when compared to the 2022 cohort. Disparities were also evident in lunchbox usage, preparation processes, consumption of self-prepared lunchboxes, and the proportion maintaining a balanced diet through these lunchboxes. These disparities were further reflected in their behaviors. Students admitted in 2021 exhibited a greater inclination to actively participate in social responsibility activities, such as purchasing socially responsible products and expressing a willingness to donate to social causes. Conversely, students admitted in 2022 displayed less enthusiasm for engaging in such activities.

The marked variations in both concepts and behaviors between students of the 2021 and 2022 academic years can be attributed to changes in the economic landscape of Taiwanese families following the COVID-19 pandemic, leading to substantial shifts in educational budgets. Public data from the Taiwan Ministry of Education during the 2021 academic year indicates that students who suspended their studies due to financial reasons accounted for 5.354% of the total number, whereas, in the 2022 academic year, this figure rose to 5.694% [4–7]. This data suggests a slower-than-expected economic recovery after the epidemic, negatively impacting job opportunities for students and the financial income of parents, thereby reducing the actual disposable amount available to students enrolling in 2022 and directly influencing their willingness to contribute to social responsibilities. However, the implementation of the "University Social Responsibility (USR) Practical Plan" by the Taiwan Ministry of Education, orchestrated by higher education institutions and guiding high school students, may have influenced the incoming students of 2021. These students experienced the impact of the COVID-19 pandemic during their high school years (2019–2021), resulting in limited opportunities to participate in USR projects led by higher education institutions. This circumstance could elucidate the willingness of 2022 academic year students to donate to social causes, while exhibiting a lower inclination to personally engage in social welfare activities.

The economic dynamics and parental financial support for students entering in 2021 and 2022 were investigated in this study. Notably, a lower percentage of students in the 2021 cohort engaged in part-time employment compared to their counterparts in 2022. Additionally, parents of students entering in 2021 were found to provide more substantial financial assistance than those of students entering in 2022. These observations are attributed to the economic repercussions of the COVID-19 pandemic on the Taiwanese economic landscape. The financial conditions of students entering in 2022 were less favorable, compounded by their parents' own financial constraints. Consequently, these students faced heightened economic challenges. When confronted with adverse economic conditions, students may show lower interest in social responsibility. However, their roles may shift, with many becoming recipients of social assistance, potentially explaining why students entering in 2021 displayed a greater willingness to donate compared to their 2022 counterparts.

Furthermore, the global economic instability induced by the COVID-19 pandemic and the resulting changes in students' lifestyles have significantly influenced their perspectives on social responsibility and public welfare activities. Prior research by Chao (2023) [10] and Bushara et al. (2023) [14] has indicated substantial changes in students' spending habits due to a reduction in disposable income during the pandemic. The post-pandemic period witnessed no significant improvement in the Taiwanese job market, leading to

increased unemployment rates among students' parents or a reduction in their income. This, in turn, affected their ability to provide financial support and allowances, resulting in a relatively lower proportion of financial support for students entering in 2022. Consequently, students from the 2022 cohort were compelled to seek part-time employment to meet their financial needs, aligning with the findings of Song et al. (2021) [15]. Despite being generally regarded as highly educated individuals in society, the financial difficulties faced by these university students may impact their willingness to purchase charitable products and make donations for public welfare, consistent with the results of this study. Students entering in 2022, experiencing heightened interest in social responsibility, exhibited a decreased inclination toward public welfare contributions compared to those entering in 2021. This suggests that, while students may express interest in social responsibility amid unfavorable economic conditions, their roles may shift, primarily making them recipients of social support. This insight may explain the observed disparity in the willingness to donate between the two cohorts.

The enrollment trends in 2022 reveal a noteworthy surge in students engaging in part-time employment compared to their counterparts in 2021. This shift could be attributed to the aftermath of the COVID-19 pandemic. An intriguing correlation emerges between the prevalence of part-time work and the financial support received from parents. Notably, 65.19% of students entering school in 2021 benefited from parental financial assistance, providing them with greater financial autonomy. In contrast, a mere 39.86% of students enrolled in 2022 received such support, compelling them to seek disposable income through part-time employment. Consequently, students entering in 2022 found themselves compelled to undertake part-time work, inevitably impacting their academic commitments. The imperative to balance work and studies often leads these students to promptly exit campus upon course completion, diverting their attention away from on-campus and extracurricular activities. Notably, the decline in participation extends to areas such as clubs, societies, and student groups, underscoring a significant departure from the patterns observed in the previous academic year. This observed phenomenon resonates with the cognitive and behavioral shifts highlighted in the preceding paragraph.

*7.2. Differences in WTP and Contingent Valuation across Different Years*

The findings of this study reveal that students enrolled in 2021 exhibit a slightly higher inclination towards donation compared to their counterparts from 2022, as evidenced by their involvement in the student-led social responsibility initiative of crafting joyful lunchboxes on campus. Subjective willingness-to-pay (WTP) observations indicate an increase of TWD 131 (approximately USD 4.30), while inferred WTP observations also show an uptick of TWD 39.88 (approximately USD 1.31). Given the heightened interest in socially responsible behaviors among the 2021 cohort, it is logical to infer that these students harbor a greater willingness to financially support campus activities, such as the student-led initiative of creating happiness lunchboxes, compared to their 2022 counterparts. These endeavors are integral to fostering sustainable campus development and realizing the vision of eradicating hunger on campus. Simultaneously, the student-led social responsibility activity aligns with the pursuit of a sustainable society and significantly contributes to the campus community.

However, the anticipated economic recovery post-COVID-19 has fallen short of expectations, leading to a deterioration in the financial circumstances of students and their parents. Consequently, there has been a decrease in the actual donation amounts from individuals genuinely willing to contribute to social welfare, despite the positive inclinations expressed by the 2021 cohort towards such initiatives. The data presented in this study highlight a notable disparity between the expressed subjective willingness to pay (WTP) and the inferred donation amounts among students enrolled in 2021, who exhibit a higher inclination to donate. Conversely, students enrolled in 2022, despite indicating a lower willingness to donate, demonstrate inferred WTP amounts surpassing their subjective expressions, revealing significant differences.

The study further reveals that students enrolled in 2021 underwent training through the Ministry of Education's University Social Responsibility (USR) program during their high school years. Consequently, their responses were influenced by social perceptions and past learning experiences, resulting in a higher self-reported donation amount and willingness to contribute. In contrast, students enrolled in 2022, when responding to questions, displayed reduced susceptibility to social perceptions and moral pressure, reflecting a lower subjective WTP. While economic factors may contribute to the decreased disposable income among students enrolled in 2022 compared to their 2021 counterparts, the inferred WTP performance is approximately TWD 50 higher than the subjective WTP. This suggests that, despite facing life pressures and economic realities, students in 2022 harbor a genuine willingness to contribute to social responsibility and public welfare, albeit being constrained in their ability to do so. This finding aligns with the objectives of Taiwan's Ministry of Education's University Social Responsibility (USR) plan, emphasizing the importance of cultivating campus social responsibility and charity activities, such as the creation of homemade happiness lunchboxes. These initiatives contribute to the overarching goal of building a sustainable society and achieving Sustainable Development Goals, underscoring the significance of engaging students in social responsibility and public welfare endeavors.

Moreover, our regression analysis underscores that both subjective willingness to pay (WTP) and inferred WTP are primarily influenced by two key factors: F2, indicating whether the surveyed students are involved in part-time employment to generate income, and F3, specifying whether their parents provide financial support for their living expenses and allowances. Notably, when students possess a source of income or enjoy greater disposable income, their inclination to contribute financially tends to be higher. This pattern aligns with findings from Foo and Tan (2021) [38] and Gaddis (2020) [39], suggesting that an increase in students' disposable income corresponds to an elevated commitment to social responsibility and charitable contributions. The campus happiness lunchboxes primarily serve functional purposes related to food and clothing, potentially falling short of meeting the expectations and needs of financially independent students. Furthermore, there exists a positive correlation between the students' interest in participating in the creation of campus happiness lunchboxes and their willingness to pay. As the enthusiasm for engaging in public welfare initiatives and giving back to society grows, so does the readiness to contribute financially. This correlation highlights the importance of aligning the nature of such initiatives with the interests and values of students, emphasizing the impact of individual engagement in public welfare undertakings on financial commitment.

In conclusion, concerning behavioral aspects, the inferred willingness-to-pay (WTP) observation reveals that the three factors—B1 (committing to my best and assuming social responsibility), B2 (intending to purchase socially responsible products), and B3 (expressing willingness to donate to social welfare)—all exhibit a significant and direct positive impact on the amount of donations inferred from WTP. This suggests that the inferred WTP group (*n* = 111) genuinely reflects the donation amounts that students, genuinely committed to participating in social responsibility and public welfare, are willing to contribute. Conversely, in the subjective WTP observation, no significant impact is observed in B1, B2, and B3 (*p* > 0.05). This implies that some respondents in the subjective WTP group (*n* = 169) may be influenced by social perceptions and past learning experiences, leading to decisions that may not truly represent their heartfelt intentions. An intriguing observation is that, when assessed from the subjective WTP group, the subjective WTP is TWD 396, whereas, from the inferred WTP group, the inferred WTP is TWD 398—indicating a close proximity between the two values. This suggests that students influenced by social perceptions and past learning experiences may collectively contribute to social responsibilities and public welfare without genuine willingness, akin to the concept of purchasing "indulgences," aligning with the findings of Cojoc and Stoian (2014) [40].

Nevertheless, the distinctiveness of this study lies in the fact that, in the realm of subjective willingness to pay (WTP), students' interest in I3 (making happiness lunchboxes on campus), I4 (having a keen interest in the cooking process), and I6 (being interested in

maintaining a balanced diet through happiness lunchboxes) significantly influences their WTP, reflecting students' consumption interests. However, in the inferred WTP, I3, I4, and I6 exhibit no significant impact. This suggests that students genuinely willing to donate are not swayed by their involvement in crafting happiness lunchboxes or their enthusiasm for the cooking process and balanced diets. Such interests do not impede their commitment to participating in social welfare initiatives. Additionally, there is no discernible impact on students' willingness to donate based on their year of enrollment, indicating that the hypothesis posited in this study—asserting consistency between students' willingness to donate and their evaluation of conditions for the same service across different enrollment years—is partially supported.

In summary, this study utilized contingent valuation to gauge students' authentic donation intentions, probing the factors influencing donation amounts and behavior. It successfully estimated the students' genuine willingness to partake in social responsibility and public welfare activities. The research also delved into advancing the United Nations' Sustainable Development Goals (SDGs), with a focus on the "Zero Hunger" and "responsible consumption and production" initiatives among the youth demographic. Future research endeavors could further explore how these factors impact contingent valuation and investigate the interplay of students utilizing contingent valuation to assess their peers' willingness to pay, both of which warrant further examination in subsequent studies. Moreover, our willingness-to-pay (WTP) results indicate that, among students enrolled in 2021, 65.93% expressed a willingness to donate in subjective WTP, whereas the inferred proportion of individuals willing to donate in WTP was 42.96%. However, for students enrolled in 2022, the percentage of those willing to donate in subjective WTP declined to 55.94%, and the inferred WTP figure further decreased to 37.06%. In summary, it is evident that students' willingness to donate is significantly influenced by the financial resources at their disposal, thereby impacting both their behavior and cognitive decisions.

*7.3. Validation of the Practical Framework*

Aligned with the previously outlined conceptual framework, a four-week validation of the practical framework was executed in this study. Initially, a preliminary pilot test was conducted within the Department of Hospitality Management. The fundraising and donation amounts for this phase totaled approximately TWD 408.38 per person (equivalent to around USD 13.6), with a cumulative contribution of TWD 15,110 (approximately USD 503.7) from 37 participants. Throughout this period, ten campus happiness lunchboxes were made available daily, with voluntary orders open to all faculty and students. The lunchbox menu was meticulously curated by faculty members specializing in food and nutrition, incorporating a comprehensive approach. Apart from ensuring fundamental food safety and hygiene standards, the menu aimed to guide consumers by providing accurate nutritional information, emphasizing the intake of calories, proteins, and fats in a healthy and responsible manner. The overarching objective was to deliver a delightful, wholesome, and secure dining experience for all faculty and students participating in the campus happiness lunchboxes initiative. The campus happiness lunchboxes were available five days a week, exclusively during lunch hours, spanning a four-week period. At the conclusion of this trial period, a total of 732 campus happiness lunchboxes were distributed, encompassing both charitable and voluntary-order lunchboxes. The donations amassed from the voluntary-order campus happiness lunchboxes reached TWD 49,105 (approximately USD 1636.8), resulting in an end-of-period surplus of TWD 15,639 (about USD 521.3) after factoring in the initial TWD 15,110 (approximately USD 503.7) in contributions. This outcome attests to the financial viability of establishing campus happiness lunchboxes on campus through student donations, enabling the execution of socially responsible and charitable initiatives at our study site.

In essence, this initiative extended assistance to approximately 200 underprivileged students across the entire campus community, irrespective of their college or department affiliation. Emphasizing accessibility, the program ensured that any student in need could

benefit without imposing conditions or obligations. As one of the pioneering studies unveiling potential factors influencing the value of inferences, the primary objective of this research was to identify elements impacting people's inferences about willingness to pay (WTP). However, within the scope of this study, the exploration into the reasons behind certain factors influencing willingness to pay was limited. Future research endeavors could delve deeper into the mechanisms guiding individuals' assessments of others' willingness to pay and explore the factors shaping inferential value.

### 8. Conclusions

This study underscores that the extrapolation of willingness to pay (WTP) can provide more reliable and conservative estimates of student giving, offering insights closer to the actual truth. By leveraging inferred WTP, the economic feasibility of student-led social responsibility and charitable activities, such as campus happiness lunchboxes funded by student donations, becomes apparent. Inferred WTP also facilitates the assessment of various factors influencing students' engagement in social responsibility and charitable donations, ultimately impacting both the actual donation amounts and willingness to donate. Furthermore, through the implementation of a small-scale practical initiative, this study affirms that utilizing conditional assessment to evaluate the economic behavior of student-led initiatives, like campus happiness lunches, is not only entirely feasible but also sustainable. Recognizing the significance of fostering sustainable campuses, universities across various Asian countries are actively making efforts to realize goals such as "Zero Hunger" and "responsible consumption and production" on campus. Despite certain factors affecting the actual donation amount, students steadfastly embrace social responsibility and charitable giving, diligently striving towards a positive and sustainable development trajectory. This ingrained concept resonates deeply within the community, aligning with its long-term mission to advance the United Nations Sustainable Development Goals (SDGs).

Drawing upon the insights gleaned from this research, several recommendations are articulated to cultivate a sustainable campus environment through social responsibility initiatives. To ensure active participation in the pursuit of a socially responsible and philanthropic campus that is sustainable, it is crucial to gauge the willingness of students, faculty, and beneficiaries to support such endeavors. Utilizing contingent valuation to assess donors' willingness and capacity to contribute, adopting a conservative approach in evaluations, and striving to minimize biases stemming from societal expectations are integral steps in conducting evaluations that closely mirror real-world scenarios. By incorporating contingent valuation, future university campuses driven by student-led social responsibility initiatives can navigate the intricate relationships among students, donors, and beneficiaries. This approach optimizes the execution of donations and activities, steering them towards the overarching goal of sustainable campus development. Embracing these recommendations enhances the likelihood of the successful implementation and long-term viability of social responsibility initiatives within the university setting.

**Author Contributions:** Z.-Y.W. contributed to the conception, the design of the study, and wrote sections of the manuscript. K.-W.C. wrote the first draft of the manuscript. All authors have read and agreed to the published version of the manuscript.

**Funding:** This research received no external funding.

**Institutional Review Board Statement:** Not applicable.

**Informed Consent Statement:** Informed consent was obtained from all subjects that were involved in the study.

**Data Availability Statement:** The original contributions presented in this study are included in the article; further inquiries can be directed to the corresponding author.

**Acknowledgments:** The authors thank the reviewers for their valuable comments.

**Conflicts of Interest:** The authors declare that they have no known competing financial interests or personal relationships that could have appeared to influence the work reported in this paper.

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
