# Peer review of "Student-Led Campus Happiness Lunchboxes: Paying for Positive Impact"

_sustainability, doi:10.3390/su16041672_

Round 1

Reviewer 1 Report

Comments and Suggestions for Authors

Within the framework of the Sustainable Development Goal of the United Nations, the Taiwanese government is taking various steps in line with these goals within the framework of zero hunger, food security and climate change themes.

In this study, the behavioural levels of university students regarding the United Nations Sustainable Development Goal and the government's actions on this issue, such as zero hunger, food security on campus, the capacity of students to donate and their behavioural patterns were examined.

In the 2nd title of the study, some of the previous researches are included and it is thought that it would be useful to develop this a little more. In addition, the study contributes to the field.

Zero hunger points to an important point in terms of showing the impact of basic problems such as food security on university campuses and the tendency of students to give aid.

Researchers should also experience the sample in other campuses in their other studies. Thus, they will be able to make comparisons and test their research.

The sources are compatible with the research and should be enriched.

In the research, 4 tables were used and the tables do not show formal compatibility. Researchers should reorganise the tables they have used.

The literature can be looked at a little more and the study should be reviewed linguistically.

Author Response

Dear Reviewer 1,

I hope this letter finds you well. I would like to express my sincere gratitude for your guidance, which has been immensely valuable to us. Your feedback has significantly contributed to refining our paper, enabling us to present our research findings more concretely, comprehensively, and accurately. Your guidance and encouragement have provided us with ample opportunities for learning and improvement, for which we are truly grateful.

In response to your constructive feedback, we have made the following revisions to the manuscript:

  1. Literature Review:

Following an extensive review of the literature, we have identified and incorporated relevant papers that align closely with our study. These include:

(1) Ho, S. S.-H., Lin, H.-C., Hsieh, C.-C., & Chen, R. J.-C. (2022). Importance and performance of SDGs perception among college students in Taiwan. Asia Pacific Education Review, 23(4), 683-693.

(2) Huang, C.-L. (2022). Innovative Community Care and the Sustainable Development of University Social Responsibility in the Post-Pandemic Era. Hu Li Za Zhi, 69(3), 4-6.

(3) Yang, J. C.-C. (2022). University and Community: Social Responsibility Practices and International Cases. Journal of Education Research, 335, 67-81.

(4) Yang, J. C.-C. (2023). University Social Responsibility in Taiwan: Diverse Goals and Interdisciplinary Learning. In Transformation of Higher Education in the Age of Society 5.0: Trends in International Higher Education. 145-157.

Additionally, we have made comprehensive revisions to the manuscript, specifically in lines 48-54, 216-227, and 543-549.

  1. Tables:

The original four tables have been reorganized, and we have verified the accuracy of the data. Corrections and improvements have been made throughout the entire manuscript in Tables 1 to 8.

We appreciate your thorough review and constructive suggestions, which have undoubtedly strengthened the quality of our paper. Please feel free to provide any further guidance or feedback you may have.

Once again, thank you for your time and invaluable support.

Sincerely,

Ze-Yung Wang

Associate Professor

Department of Hospitality Management

Tajen University

No.20, Weixin Rd., Yanpu Township, Pingtung County 90741, Taiwan,

Telephone: +886 982483888

Email: zyrudder@gmail.com

Reviewer 2 Report

Comments and Suggestions for Authors

Thank You for the opportunity to review the manuscript titled: " Student-Led Campus Happiness Lunchboxes: Paying for Positive Impact" for Sustainability.

The manuscript in question provides an interesting insight into behaviors of students and their commitment to social responsibility.

Overall, this was an enjoyable read of utmost importance for health and many other policy professionals involved with students’ welfare and wellbeing. It is relatively cohesively written.

I only have a couple of suggestions for the respectable authors.

Lines 33-35: Food waste is the subject of SDG 12, not SDG 2 as is being presented in the manuscript.

Lines 158-159: Why was the one year difference considered sufficient to efficiently reveal the expected changes?

Lines 196-197: Was the questionnaire constructed especially for this study? How was it constructed, tested…? The overall impression is that the process of construction of the questionnaire was not sufficiently described. What were the basis used for its construction, scientific background, any other surveys used as models…?

Sincerely,

The reviewer

Author Response

Dear Reviewer 2

I hope this letter finds you well. I would like to express my sincere gratitude for your guidance, which has been immensely valuable to us. Your feedback has significantly contributed to refining our paper, enabling us to present our research findings more concretely, comprehensively, and accurately. Your guidance and encouragement have provided us with ample opportunities for learning and improvement, for which we are truly grateful.

In response to your valuable feedback, we have implemented the following revisions to the manuscript:

  1. Lines 33-35:

We appreciate your observation regarding the misclassification of SDG 12 in our manuscript. Following a review of the regulations outlined in the Taiwan Ministry of Education's USR project guidelines, we have conducted a literature review focusing on SDG 12. The necessary corrections have been made in the manuscript, specifically in lines 66-72 and 211-219.

  1. Lines 158-159:

Thank you for bringing attention to the oversight in lines 158-159. We apologize for the omission. The one-year difference in the study population was not adequately explained in the original manuscript. Students admitted in 2021 had a full year of exposure to USR initiatives and feedback during their first year of high school. In contrast, students admitted in 2022 experienced disruptions due to COVID-19, with limited exposure to USR initiatives during their high school years. The manuscript has been amended to address this oversight, specifically in lines 287-309.

  1. Lines 196-197:

We appreciate your inquiry regarding the construction of the questionnaire. We acknowledge the oversight in the original manuscript. The questionnaire design was informed by existing academic theories and literature, ensuring representative coverage. The questionnaire underwent expert review by three representatives (academic, social welfare industry, and government social welfare officials). It further underwent item and factor analysis to confirm dimensions, followed by a pre-test to ensure reliability and validity. The manuscript has been revised to provide a more thorough description, specifically in lines 310-321.

We sincerely thank you for your meticulous review and valuable suggestions, which have undoubtedly strengthened the quality of our manuscript. If you have any additional guidance or feedback, please do not hesitate to share.

Once again, thank you for your time and invaluable support.

Sincerely,

Ze-Yung Wang

Associate Professor

Department of Hospitality Management

Tajen University

No.20, Weixin Rd., Yanpu Township, Pingtung County 90741, Taiwan,

Telephone: +886 982483888

Email: zyrudder@gmail.com

Reviewer 3 Report

Comments and Suggestions for Authors

Line 69 - note the contradiction in the expression "addressing zero hunger" 

Line 370 to 399 - The discussion presents the results of the study.

Line 400 to 413 - The paragraph presents the justification for the study and should be in the introduction.

The discussion is weak and repeats the data presented in the results.

The conclusion is not clear about the objectives presented in the study.

Author Response

Dear Reviewer 3

I hope this letter finds you well. I would like to express my sincere gratitude for your guidance, which has been immensely valuable to us. Your feedback has significantly contributed to refining our paper, enabling us to present our research findings more concretely, comprehensively, and accurately. Your guidance and encouragement have provided us with ample opportunities for learning and improvement, for which we are truly grateful.

In response to your valuable feedback, we have implemented the following revisions to the manuscript:

  1. Line 69:

We appreciate your guidance on the contradiction in the expression "addressing zero hunger" in the Introduction. The introduction has undergone a comprehensive review and revision to enhance clarity and accuracy of semantic expression. The manuscript has been revised from line 48 to line 107.

  1. Lines 370 to 399:

Thank you for pointing out the presentation of study results in the discussion. The discussion section has been completely rewritten, strengthening the presentation of research findings.

  1. Lines 400 to 413:

 This has been rectified.

  1. Discussion and Conclusion:

Your observation regarding the weakness of the discussion, repetition of data in the results, and the lack of clarity in the conclusion regarding the study objectives is duly noted. The Results, Discussion, and Conclusion sections have all been rewritten to strengthen the presentation of research outcomes. I appreciate the opportunity for improvement and learning that your guidance has provided.

Once again, thank you for your time and invaluable support.

Sincerely,

Ze-Yung Wang

Associate Professor

Department of Hospitality Management

Tajen University

No.20, Weixin Rd., Yanpu Township, Pingtung County 90741, Taiwan,

Telephone: +886 982483888

Email: zyrudder@gmail.com

Round 2

Reviewer 3 Report

Comments and Suggestions for Authors

The article is adequately for publication.